

# A Possible Unaccounted Source of Nitrogen-Containing Compounds Formation in Aerosols: Amines Reacting with Secondary Ozonides

Junting Qiu[1,2,3], Xinlin Shen[1,2], Jiangyao Chen[1,2], Guiying Li[1,2], Taicheng An[1,2]

[1]Guangdong-Hong Kong-Macao Joint Laboratory for Contaminants Exposure and Health, Guangdong Key Laboratory of Environmental Catalysis and Health Risk Control, Institute of Environmental Health and Pollution control, Guangdong University of Technology, Guangzhou 510006, China.
[2]Guangzhou Key Laboratory of Environmental Catalysis and Pollution Control, Key Laboratory of City Cluster Environmental Safety and Green Development of the Ministry of Education, School of Environmental Science and Engineering, Guangdong University of Technology, Guangzhou 510006, China.
[3]Graduate School of Frontier Sciences, The University of Tokyo, 5-1-5 Kashiwanoha, Kashiwa 277-8563, Japan.

*Correspondence to*: Taicheng An (antc99@gdut.edu.cn)

**Abstract.** Nitrogen (N)-containing compounds have a significant impact on the optical and toxicological properties of aerosols. 1,2,4-Trioxolanes, known as secondary ozonides (SOZs), key products from the ozonolysis of biogenic terpenoids, are readily taken up into atmospheric aerosols and act as oxidants, potentially interacting with amines in the atmosphere. In the present work, we carefully investigated the component of the particles produced by the ozonolysis of β-caryophyllene (β-C) in the presence of ethylamine (EA), methylamine (MA), dimethylamine (DMA) or ammonia. The mass spectrometric results show that SOZ is the dominant product from the ozonolysis of β-C. It readily reacts with EA and MA but has inert reactivities toward DMA and ammonia. Similar experimental results were achieved with α-humulene (α-H), an isomer of β-C, was used in place of β-C. Additionally, $D_2O$ and $H_2^{18}O$ solvents were used for the characterization of products. The results revealed an intriguing phenomenon that the products from β-C SOZ and α-H SOZ reacting with the same amine (EA or MA) possessed different functional groups, despite the fact that they are isomerized species with the identical chemical structure (1,2,4-trioxolane). That indicates the chemical conformation of SOZs has a strong influence on how they react with amines. For the first time, SOZs derived from β-C and α-H reacting with amines is reported in this study, that may represent a hitherto unrecognized source of N-containing compounds production in atmospheric aerosols.

## 1 Introduction

Nitrogen (N)-containing compounds are ubiquitous in atmospheric aerosols, which are of great importance in acting as light-absorbing species in aerosols (Laskin et al., 2015), meanwhile, some N-containing compounds are considered to be hazardous to human health (Albinet et al., 2008). A main contributor to the formation of N-containing compounds is the progress that $NO/NO_2$ are transferred to terminal products in $NO_x$ radical cycle, including peroxy nitrates ($RO_2NO_2$), alkyl nitrates ($RONO_2$), and nitric acid ($HNO_3$) (Perring et al., 2013). Additional route for N-containing species formation in the atmosphere is the oxidation of VOCs by the nitrate radical ($NO_3$) (Ng et al., 2017; Fry et al., 2014). On the other hand,



atmospheric amine chemistry, involving new particle formation (NPF) events is another important contributor to N-containing compounds in aerosols. The substitution by one or more organic functional groups leads to stronger basicity of amines than ammonia, indicating that amines are readily to participate in NPF though acid-base reactions, which have been

confirmed in numerous field researches (Yao et al., 2016; Yu et al., 2012) and laboratory works (Almeida et al., 2013; Erupe et al., 2011; Glasoe et al., 2015; Kurten et al., 2014; Tong et al., 2020).

  Amines are massively emitted from various biogenic and anthropogenic sources, such as biomass burning, animal husbandry, ocean organisms, automobiles and industries (Ge et al., 2011). Aliphatic amines with low molecular weight, including methylamine (MA), dimethylamine (DMA), trimethylamine (TMA), or ethylamine (EA), are dominant species of

amines in the atmosphere. Yao et al. reported a high concentration of total DMA and EA up to 130 pptv in Shanghai, China (Yao et al., 2016). In the boreal forest, the concentration of total DMA and EA were detected to be around 150 pptv (Kieloaho et al., 2013). Amines involved atmospheric chemistry regarding the formation of N-containing compounds and growth of secondary organic aerosols (SOAs) has been investigated in cohort works. For instance, amines are found to be fairly reactive toward important atmospheric aldehydes in condense phase, such as glyoxal, methylglyoxal, glycoaldehyde,

or acetaldehyde, which significantly affect the physiochemical properties of aerosols and contribute to SOAs growth (De Haan et al., 2011; Galloway et al., 2014). A recent study shows that alkylaminium carboxylates formed from the reactions of amines with organic acids have lower vapor pressures than original organic acids, implying that alkylaminium carboxylates could enhance SOAs formation (Lavi et al., 2015). Another study carried out by the same group reinforced the crucial role of alkylaminium carboxylates in determining the characteristics of aerosols, for the reason that alkylaminium carboxylates are

capable of enhancing the particle hygroscopicity and the cloud condensation nuclei activity (Gomez-Hernandez et al., 2016). In addition, Duporte et al. reported a systematic research on the ozonolysis of α-pinene in the presence of DMA, and they found that DMA was able to react with aldehydes or carboxylate acids generated from the ozonolysis of α-pinene, enhancing SOAs formation (Duporte et al., 2017).

  Amines can also be directly oxidized by major atmospheric oxidants, such as OH, $O_3$ and $NO_3$, in gas phase or in

condensed phase (Ge et al., 2011; Qiu and Zhang, 2013; Tang et al., 2013). Recently, Criegee intermediates reacting with amines has been investigated in some studies, however, the actual effect of Criegee intermediates on oxidizing amines is by now unclear (Chhantyal-Pun et al., 2019b; Kumar and Francisco, 2019; Ma et al., 2020; Mull et al., 2020). 1,2,4-Trioxolanes, known as secondary ozonides (SOZs), are formed by the intramolecular reactions of the Criegee moieties with the carbonyl endogroups, as well as bimolecular reaction of Criegee intermediates with carbonyl species such as formaldehyde and

acetone (Chhantyal-Pun et al., 2020; Chhantyal-Pun et al., 2019a; Cornwell et al., 2021; Wang et al., 2022). SOZs are major products from the ozonolysis of important biogenic terpenoids, such as limonene, carene, β-pinene, β-caryophyllene and α-humulene (Vibenholt et al., 2009; Winterhalter et al., 2000; Nguyen et al., 2009; Winterhalter et al., 2009; Beck et al., 2011), and they are readily taken up into atmospheric aerosols (Yao et al., 2014). The formation of SOZs occurs not only in the gas phase but also in bulk liquid phases (Griesbaum et al., 1996) and at gas–liquid/solid interfaces (Enami et al., 2008;

Karagulian et al., 2008; Coffaro and Weisel, 2022). Additionally, SOZs can also be formed via OH reactions of lipid





molecules (Zeng et al., 2020; Zhang et al., 2018). Since SOZs are categorized as both organic peroxides and reactive oxygen species (Sanchez and Myers, 2000), they potentially function as oxidants and interact with amines in atmospheric condensed phase. Therefore, the aim of this study is to determine whether the interaction of SOZs with amines results in the formation of N-containing compounds.

β-Caryophyllene (β-C) and α-humulene (α-H) are representative sesquiterpenes (Arey et al., 1995; Helmig et al., 2007), and their chemical structures are listed in Figure S1. Albeit not predominant terpene species like isoprene or α-pinene, β-C and α-H are of special significance as powerful SOA makers, due to their rapid degradation in the atmosphere and the low volatility of the degradation products. SOZs are dominant products from the ozonolysis of both β-C and α-H (Nguyen et al., 2009; Winterhalter et al., 2009; Beck et al., 2011), thus, in this study, we choose the reactions of β-C and α-H with $O_3$ to

produce SOZs, and investigate the reactivities of SOZs toward amines. Firstly, we carefully carry out the ozonolysis experiments of β-C in the absence/presence of EA in a smog chamber. The particles generated inside the smog chamber are monitored and the chemical components of the particles are detected by mass spectrometry. $D_2O$ and $H_2^{18}O$ isotope labelling experiments were performed for the identification of the products detected. Next, chemical structures of the products formed from EA reacting with β-C SOZ and α-H SOZ are compared to know the effect of molecular conformation on the reaction

mechanism of SOZs. In addition, the reactivities of SOZs toward MA, EA, DMA or ammonia are also comparably investigated.

## 2 Experimental

### 2.1 Ozonolysis experiment

All the ozonolysis experiments were performed in a smog chamber. Since the details of the smog chamber has been reported

in other articles (Luo et al., 2021; Luo et al., 2020), we just make a brief description herein. Two extremely same pillow-shaped Teflon reactors (2.5 m × 2.0 m) mounted inside the smog chamber, and each reactor is surrounded by 3 high efficiency ionizing blower (varied from 0 to a maximum of 2000 r/min), in order to mix the air inside the reactors evenly. All the experiments were carried out in one reactor under zero air background at a volume of 1000 L, with no detectable particles, < 0.5 ppb non-methane hydrocarbon (NMHC), and < 1 ppb $NO_x$, $O_3$ and carbonyl compounds. The relative

humidity inside Teflon reactors is (RH) ≤ 5%, and the temperature is kept at 295 ± 3 K.

Schematic experimental procedure for the ozonolysis of β-C in the presence of EA is presented in Figure S2. β-C is well mixed with EA before the addition of $O_3$, and the initial concentrations of chemical species inside a reactor are β-C 200 ppb, EA 80 ppb, $O_3$ 50 ppb. Other experiments (such as 200 ppb β-C+ 80 ppb MA + 50ppb $O_3$, or 200 ppb α-H + 80 ppb ammonia + 50 ppb $O_3$), are carried out with the same method. Because the reaction rate of $O_3$ toward β-C (1.4 × $10^{-14}$ $cm^3$

molecule$^{-1}$ s$^{-1}$) or α-H (1.2 × $10^{-14}$ $cm^3$ molecule$^{-1}$ s$^{-1}$) (Atkinson and Arey, 2003) is much faster than amines or ammonia ($10^{-18}$ – $10^{-21}$ $cm^3$ molecule$^{-1}$ s$^{-1}$) (Ge et al., 2011) and the initial concentration of β-C/α-H (200 ppb) is times of $O_3$ (50 ppb), $O_3$ will almost be consumed intermediately via the reaction with β-C/α-H after being injected inside the reactor. The products



from ozonolysis of β-C/α-H may subsequently participate in the reactions with amines or ammonia. Ozonolysis leads to the generation of particle matters via condensation of oxidized low-volatility species, which are able to be monitored by a scanning mobility particle sizer (SMPS, TSI). All experiments were performed in dark conditions and without an OH scavenger.

## 2.2 Particle collection and analysis

Particles with considerable sizes were collected on 47 mm quartz filters at a timing of 3 h after the injection of ozone, and quartz filters were pretreated via 8 h baking inside a Muffle furnace at a temperature of 450 °C. All the filter samples were wrapped in aluminum foil and stored in a freezer at -18 °C until extraction. Extraction of particle phase compounds was performed by soaking filter samples in 5 mL mixture of acetonitrile/ultrapure water (AN/W, vol/vol = 4/1) for 30 min at room temperature. $D_2O$ and $H_2^{18}O$ were also used in extraction instead of ultrapure water for a detailed characterization of products. A high-resolution electrospray ionization mass spectrometer (ESI-MS, ThermoFisher Q Extract quadrupole-Orbitrap) was applied in the detection of chemical compounds extracted in solutions.

## 2.3 Materials

Gas phase chemicals, such as $O_3$ and MA, were directly injected into the reactor, while liquid phase chemicals, such as EA and DMA, were injected slowly through a T-junction connected to a Fluorinated Ethylene Propylene line and spread with the flow of purified dry air, by using airtight syringes (Shanghai Anting). $O_3$ was generated by a commercial ozone generator, and the amount of $O_3$ was carefully calculated according to the injection time and the power of the ozone generator. Meanwhile, the concentrations of $O_3$ inside the reactor were confirmed by an $O_3$ analyzer (Model 49i, Thermo Scientific) before the operation of experiments. Ultrapure water was obtained from a Millipore Milli-Q water purification system (Xiamen Research Water Purification Technology, Unique-R20, resistivity ≥18.2 MΩ cm at 298 K). Chemicals β-caryophyllene (Tokyo Chemical Industrial, > 95%), α-humulene (Tokyo Chemical Industrial, > 93%), methylamine (Wuhan Newradar, 98.6 ppm mixed in $N_2$ gas), ethylamine (Aladdin Industrial, 70 wt % in $H_2O$), dimethylamine (Aladdin Industrial, 40 wt % in $H_2O$), ammonia solution (Aladdin Industrial, 25 wt % in $H_2O$), acetonitrile (Aladdin Industrial, ≥ 99.9%), $D_2O$ (J&K, > 99.8 atom % D) and $H_2^{18}O$ (Macklin, > 97 atom % $^{18}O$) were used as received.

## 3 Results and discussion

### 3.1 Reaction of β-C SOZ with EA

The particle formation was monitored by SMPS from the ozonolysis of β-C in the absence/presence of EA, and the chemical components of the particles were analyzed, as shown in Figure 1. Because β-C is fairly reactive toward $O_3$ and the products generated in situ have extremely low volatility, the formation of particles can be observed intermediately after $O_3$ being injected inside the Teflon reactor. The number of particles decreased rapidly as presented in Figure 1A due to their



coagulation to form larger particles or deposition on the wall. According to Figure 1B, the total particle volume produced by the ozonolysis of β-C grew initially before beginning to decline about 10 minutes after the particle loss rate surpassed the

creation rate. With the addition of EA, no discernible change in the number concentration of particles was seen in Figure 1A, however the volume of total particles slightly increased as shown in Figure 1B.

The addition of EA has limited effect on promoting particle formation, on account of the volatility of the products from the ozonolysis of β-C is sufficiently low. Nevertheless, the reaction of EA with the products from the ozonolysis of β-C considerably altered the chemical components of particles as shown in Figure 1C that contrasts the positive-ion ESI mass

spectra of products from the ozonolysis of β-C in the absence/presence of EA respectively. In the β-C + $O_3$ experiment, the prominent signal rise at m/z 275 that is assigned to $Na^+$-adducted $C_{15}H_{24}O_3$ species, 275 = 252 ($C_{15}H_{24}O_3$) + 23 ($Na^+$). Both experimental and theoretical research have explicitly explored the mechanisms on the ozonolysis of β-C (Nguyen et al., 2009; Winterhalter et al., 2009). Major species of $C_{15}H_{24}O_3$ are SOZ, vinyl ROOH and carboxylic acid, which are isomerized products of Criegee intermediate as presented in Scheme 1. Furthermore, product appeared at m/z 275 was identified by the

method of replacing the AN/W with AN/$D_2O$ (vol/vol = 4/1) in the extraction process. As a result, m/z 275 has no mass-shift as presented in Figure S3, implying that the product appeared at m/z 275 should be $[SOZ + Na]^+$, since it possesses no exchangeable H-atom. In contrast, the H-atoms of vinyl ROOH and carboxylic acid are exchangeable with D-atom.

The observation that intensity of m/z 275 clearly decreased in the experiment of β-C + EA + $O_3$ indicates that β-C SOZ readily react with EA. Intense peaks appear at m/z 280 (**P1**), which are assigned to the $H^+$-adducted products from β-C SOZ

reacting with EA, 280 ($C_{17}H_{29}O_2N + H^+$) = 252 ($C_{15}H_{24}O_3$) + 45 ($C_2H_7N$) − 18 ($H_2O$) + 1 ($H^+$). The vast majority of P1 is formed from the heterogeneous reaction of EA with SOZ in condensed phase. The evidence can be found in Figure S4, that is even when EA is added in the reactor 30 min after the start of ozonolysis (a situation that SOZ is almost in condensed phase), more than half of the P1 is still produced compared to the situation that EA is well mixed.

The possibility that EA vapor condensed on the particles first and then react with β-C SOZ in the extraction process was

ruled out, by directly adding EA into β-C SOZ dissolved solution. As a result, no signal appeared at m/z 280, confirming the fact that the reaction of β-C SOZ with EA took place in the smog chamber.

To the best of our knowledge, the phenomenon that β-C SOZ reacting with EA leads to the production of N-containing compounds, was reported for the first time in this work. Next, chemical analysis of products appearing at m/z 280 was conducted to better comprehend the chemical structures of previously unreported N-containing compounds.

### 155 3.2 Chemical identification of P1

$D_2O$ and $H_2^{18}O$ isotope labelling experiments were performed for the chemical identification of P1 in this work. This method was proven to be beneficial for examining the chemical structure of unidentified species in previous studies (Qiu et al., 2019; Qiu et al., 2020b; Qiu et al., 2020a). Figure 2 shows high-resolution positive-ion ESI mass spectra of products extracted in AN/W (vol/vol = 4/1), AN/$D_2O$ (vol/vol = 4/1) and AN/$H_2^{18}O$ (vol/vol = 4/1) solutions, from the reaction of β-C SOZ with

EA, where P1 ($C_{17}H_{29}O_3N + H^+$) appeared at m/z 280. 226 in the AN/W experiment. P1 shifted by + 3 mass units in the



AN/D$_2$O experiment and + 2 mass units in the AN/H$_2$$^{18}$O experiment, showing that P1 possess two exchangeable H-atoms (H$^+$ → D$^+$ contributed another + 1 mass unit) and one exchangeable O-atom, respectively.

The mechanism of β-C SOZ reacting with EA can be explained as follow. The electronegativity of the neighboring oxygens induced a net positive charge on the α-carbon of β-C SOZ. EA acting as a nucleophile may add to α-carbon and

cleave β-C SOZ. This theory is supported by a previously reported research by Na et al., which revealed that ammonia reacts with styrene SOZ via a nucleophilic attack at the α-carbon of styrene SOZ, producing benzaldehyde, hydrogen peroxide, and phenylmethanimine in the process (Na et al., 2006). Despite the attack of EA opened the cyclic structure of β-C SOZ, we didn't detect cleavage products as Na et al. reported. Instead, P1 detected in this work is the product from the addition reaction between β-C SOZ and EA, and a water molecule was removed in the process. Based on the molecular weight of P1

and the results of D$_2$O and H$_2$$^{18}$O isotope labelling experiments, a potential structure of P1 is presented in Scheme 2. It has two active H-atoms in the -NH and -OH moieties, and C=$^{16}$O can be transferred into C=$^{18}$O via H$_2$$^{18}$O addition reaction (see Scheme S1 in SI).

### 3.3 Reactions of β-C SOZ with MA, DMA and Ammonia

By replacing EA with MA, DMA or ammonia, another three smog chamber experiments were carried out. The particles were

sampled and analyzed by electrospray mass spectrometry as presented in Figure 3. Obviously, in the β-C + MA + O$_3$ experiment, intensity of m/z 275 clearly diminished and intense peak appeared at m/z 266 (**P2**), which is assigned to the H$^+$-adducted product from β-C SOZ reacting with MA, 266 (C$_{16}$H$_{27}$O$_2$N + H$^+$) = 252 (C$_{15}$H$_{24}$O$_3$) + 31 (CH$_5$N) – 18 (H$_2$O) + 1 (H$^+$). In sharp contrast, the intensities of m/z 275 in the other two experiments are essentially identical to that in the β-C + O$_3$ experiment, suggesting that DMA and ammonia have inert reactivities toward β-C SOZ. The results of D$_2$O and H$_2$$^{18}$O

isotope labelling experiments of P2 shown in Figure S5A are similar to those of P1, which indicates P2 and P1 formed through the same process. Scheme S2 presents a potential structure of P2.

Substituted by one alkyl moiety, EA or MA are considered more basic than ammonia, that potentially increased the reactivity of EA or MA toward β-C SOZ. On the other hand, the reason why DMA is less reactive than EA and MA can be explained that DMA possesses two alkyl moieties, resulting in a steric hindrance that would limit the accessibility of DMA

to β-C SOZ. Na et al. also pointed out α-methylstyrene SOZ is less reactive than styrene SOZ toward ammonia, due to being sterically hindered by the methyl group attached to the α-carbon of 1,2,4-trioxolane (Na et al., 2006). Moreover, in order to obtain more information about the mechanism of SOZs reacting with amines, in the following section, we mainly report the reactions of another SOZ produced by the ozonolysis of α-humulene (α-H), an isomer of β-C, with EA, MA DMA and ammonia.

### 190 3.4 Reactions of α-H SOZ with EA, MA, DMA and Ammonia

The smog chamber experiments of α-H + amines/ammonia + O$_3$ were performed by using the same procedure, and the chemical composition of particles generated in the reactor were analyzed by the mass spectrometer as shown in Figure 4.



SOZ (m/z 275, $C_{15}H_{24}O_3$ + Na$^+$) is the dominant product from the ozonolysis of α-H, which is consistent with the previous study (Beck et al., 2011). As can be seen, the behavior of α-H SOZ is comparable to that observed in the experiments of β-C,

exhibiting inert reactivities toward DMA and ammonia and selectively reacting with EA and MA. The products from α-H SOZ reacting with EA and MA appeared at m/z 280 (**P3**) and 266 (**P4**). Since α-H SOZ and β-C SOZ are isomerized species, molecular formula of P3 and P4 should be same to P1 and P2 respectively, which are 280 ($C_{17}H_{29}O_2N$ + H$^+$) = 252 ($C_{15}H_{24}O_3$) + 45 ($C_2H_7N$) – 18 ($H_2O$) + 1 (H$^+$) and 266 ($C_{16}H_{27}O_2N$ + H$^+$) = 252 ($C_{15}H_{24}O_3$) + 31 ($CH_5N$) – 18 ($H_2O$) + 1 (H$^+$).

D$_2$O and H$_2$$^{18}$O isotope labelling experiments were also performed for chemical identification of P3 and P4. High-resolution positive-ion ESI mass spectra of P3 extracted in AN/W (vol/vol = 4/1), AN/D$_2$O (vol/vol = 4/1) and AN/H$_2$$^{18}$O (vol/vol = 4/1) solutions are demonstrated in Figure 5. The observation that P3 shifted by + 2 mass units in AN/D$_2$O experiment and has no mass-shift in AN/H$_2$$^{18}$O experiment indicates that P3 generated from α-H SOZ reacting with EA possesses only one exchangeable H-atom and no exchangeable O-atom. A probable contributor to the exchangeable H-atom is the -NH moiety. Since P3 possesses no carbonyl or hydroxyl moieties, a dioxirane structure generated by the breaking of

C-O bonds appears to be plausible for P3, and a potential structure of P3 is deducted and displayed in Scheme 3. The results from isotope labelling experiments of P4 are presented in Figure S5B, and they point out a production mechanism of P4 that is similar to that of P3 as illustrated in Scheme S2. What should be mentioned is that α-H contains three endocyclic double bonds which are able to be attacked by ozone to generate different SOZs. As a result, P3 and P4 may have multiple

conformations, however, to simplify the representations, just one kind is provided here as an example of each. In addition, it is worth noting that the signal intensity of P4 in the α-H + MA + O$_3$ experiment is rather weak, even though the majority of α-H SOZ has been consumed via its reaction with MA and contribute to the formation of P4. This phenomenon can be explained that dioxiranes are active species and their stabilities are highly dependent on their molecular structures (El-Assaad et al., 2022). In other words, P4 dissipated rapidly after it formed because it is less stable than P3.

An astonishing phenomenon revealed here is that the product of α-H SOZ reacting with EA (P3) bear no resemblance to that of β-C SOZ reacting with EA (P1), in spite of the fact that α-H SOZ is an isomerized species of β-C SOZ and they share a same chemical structure of 1,2,4-trioxolane, which suggests that the molecular conformations of SOZs have a substantial impact on their reaction mechanism, resulting in the formation of N-containing products processing various functional groups.

**4 Atmospheric implications**

The predominant source of SOZs is biogenic terpenoids, which are emitted to atmosphere at a rate of $10^{14}$ g/year.(Guenther et al., 1995) SOZs originating from terpenoids are less volatile and hence more easily taken up into aerosols.(Yao et al., 2014) Multiphase ozonolysis and OH oxidations of unsaturated organic compounds possessing C=C bond(s) also produce SOZs, which causes an accumulation of SOZs in condensed phases (Heine et al., 2017; Enami et al., 2008; Karagulian et al., 2008;





Coffaro and Weisel, 2022; Zhou et al., 2022; Zhang et al., 2018). Terpenoid-derived SOZs are comparatively stable organic peroxides in condensed phase. For instance, A recent study revealed that $C_{10}$ and $C_{13}$ SOZs derived from terpineol can persist in water for weeks (Qiu et al., 2022). In addition, products from the ozonolysis of terpenoids including SOZs, are usually surface active in aerosols (Qiu et al., 2018b; Qiu et al., 2018a), which facilitated their reactions toward gas-phase amines. Thus, SOZs reacting with amines is probably a nonnegligible source of N-containing compounds formation in aerosols.

Moreover, Na et al. reported that SOZs derived from styrene and α-methylstyrene can react with ammonia (Na et al., 2006). In sharp contrast, our research suggests that both β-C SOZ and α-H SOZ exhibit inert reactivities toward ammonia but readily react with EA and MA. Additionally, we discovered that SOZs in different conformations reacting with EA or MA produces N-containing compounds with various functional groups. The aforementioned works indicate that the interaction of SOZs with amines or ammonia is a complicated process in the real atmosphere, leading to the formation of various N-

containing compounds. Due to the distinct roles that N-containing compounds with different functional groups play in the properties of aerosols (Laskin et al., 2015), a thorough investigation on the mechanism of SOZs reacting with amines is still urgently required. Nevertheless, the information obtained in the present study is just the tip of an iceberg, and detailed laboratory works combined with field researches are necessary toward a full picture of N-containing compounds originating from SOZs reacting with amines or ammonia.

Apart from SOZs, other organic peroxides like ROOH or ROOR', play more significant roles in atmospheric chemistry (Wang et al., 2023). For example, the oxidation of dissolved $SO_2$ by organic peroxides has been considered as a main source of sulfate formation in aerosols (Dovrou et al., 2021; Dovrou et al., 2019; Wang et al., 2021; Wang et al., 2019; Yao et al., 2019). In addition, organic peroxides can directly interact with transitional metal ions via Fenton-like reactions mechanism (Fang et al., 2020; Hu et al., 2021; Tong et al., 2016; Wei et al., 2022). However, the interaction of organic peroxides with amines has rarely been reported in previous studies. The present study recommends extensive research on organic peroxides

including SOZs reacting with amines, which will deepen our understanding of the source of N-containing compounds and benefit in the works on precisely evaluating the effects of atmospheric aerosols on human health and climate (Seinfeld et al., 2016; Shiraiwa et al., 2017; Shrivastava et al., 2017).

**5 Conclusion**

In this study, the chamber experiments showed that the component of particles produced by the ozonolysis of both β-C and α-H was dramatically altered in addition of EA or MA, which was originated from the reactions of SOZs with EA or MA. However, both β-C SOZ and α-H SOZ were found have inert reactivities toward DMA and ammonia. Additionally, $D_2O$ and $H_2^{18}O$ isotope labelling experiments revealed that the products from β-C SOZ and α-H SOZ reacting with the same amine (EA or MA) possessed different functional groups, despite β-C SOZ and α-H SOZ are isomerized species and share a same

chemical structure of 1,2,4-trioxolane. The experimental results obtained in this study indicate that a variety of N-containing



compounds can be generated via the interaction of SOZs with amines, which may constitute a hitherto unaccounted source of N-containing compounds formation in atmospheric aerosols.

*Data availability.* The data that support the results are available upon request. Please email Junting Qiu (paziqjt@gamil.com)


*Supplement.* The supplement related to this article is available online.

*Author contributions.* TA and JQ designed research. JQ and XS performed experiments. JQ analyzed the data. All the authors participated in writing the paper.


*Competing interests.* The authors declare that they have no conflict of interest.

*Financial support.* This work is financially supported by a National Natural Science Foundation of China (42020104001, 42107118 and 42177354), Local Innovative and Research Teams Project of Guangdong Pearl River Talents Program
(2017BT01Z032), and China Postdoctoral Science Foundation (2021M700881).

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





**Figure 1 Effects of EA on size and chemical composition of particles produced by the ozonolysis of β-C. (A) Number**
**concentrations of particles, (B) Volume concentrations of total particles, (C) Positive-ion ESI mass spectra of the chemical**
**components of particles extracted in AN/W (vol/vol = 4/1) solutions. Blue dots and red dots represent three independent**
**experiments**



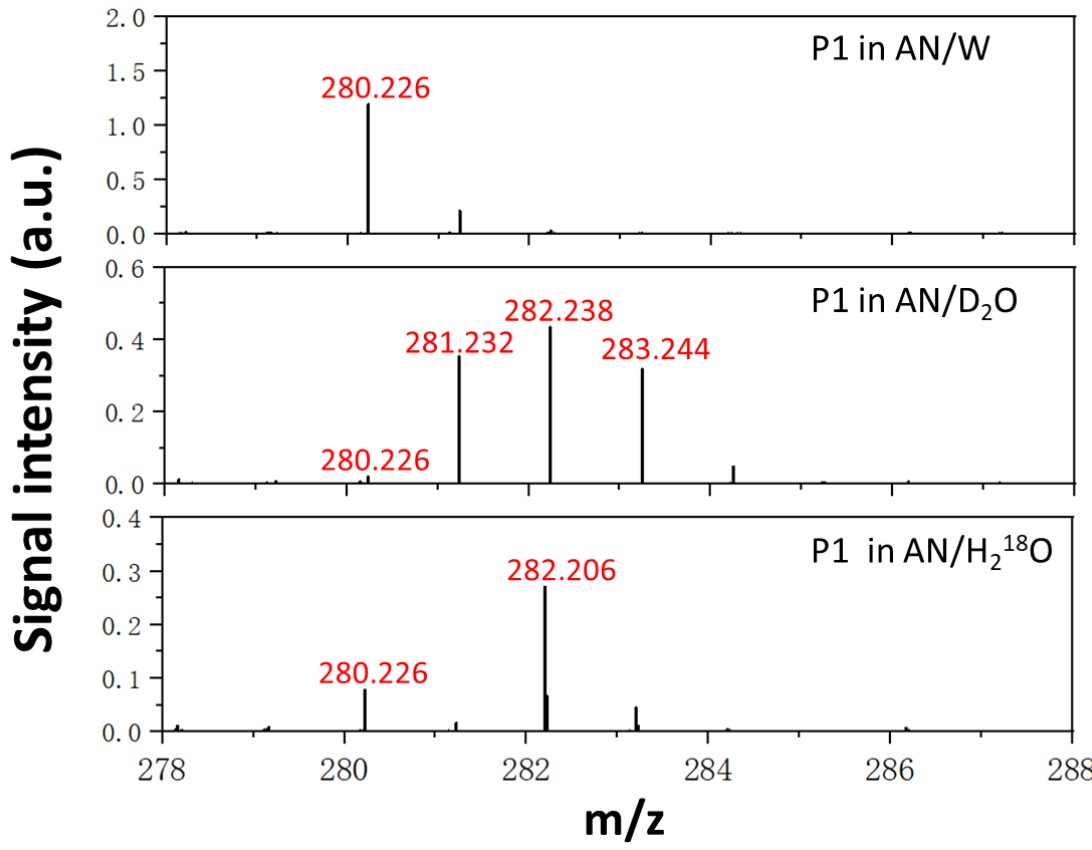

**Figure 2 High-resolution positive-ion ESI mass spectra of P1 extracted in AN/W (vol/vol = 4/1), AN/D₂O (vol/vol = 4/1) and AN/H₂¹⁸O (vol/vol = 4/1) solutions**



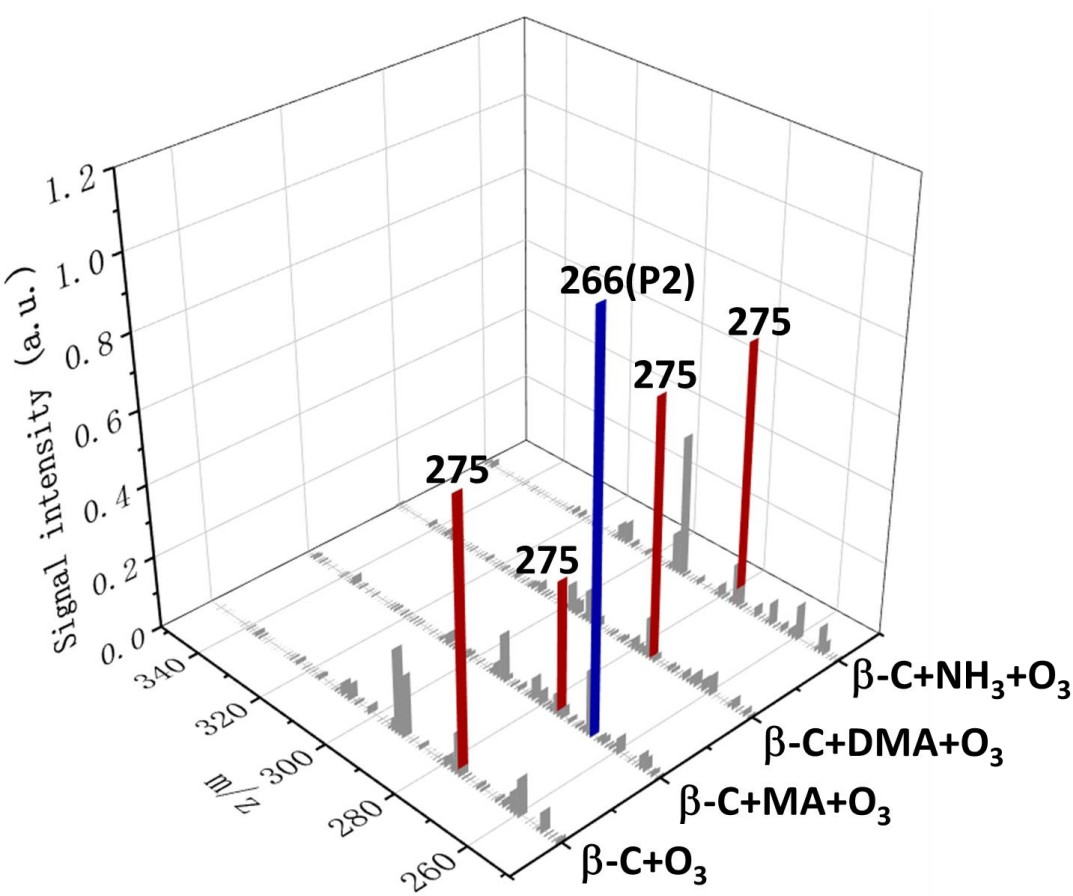

**Figure 3 Positive-ion ESI mass spectra of the products extracted in AN/W (vol/vol = 4/1) from ozonolysis of β-C in the**
**absence/presence of MA, DMA, or ammonia**



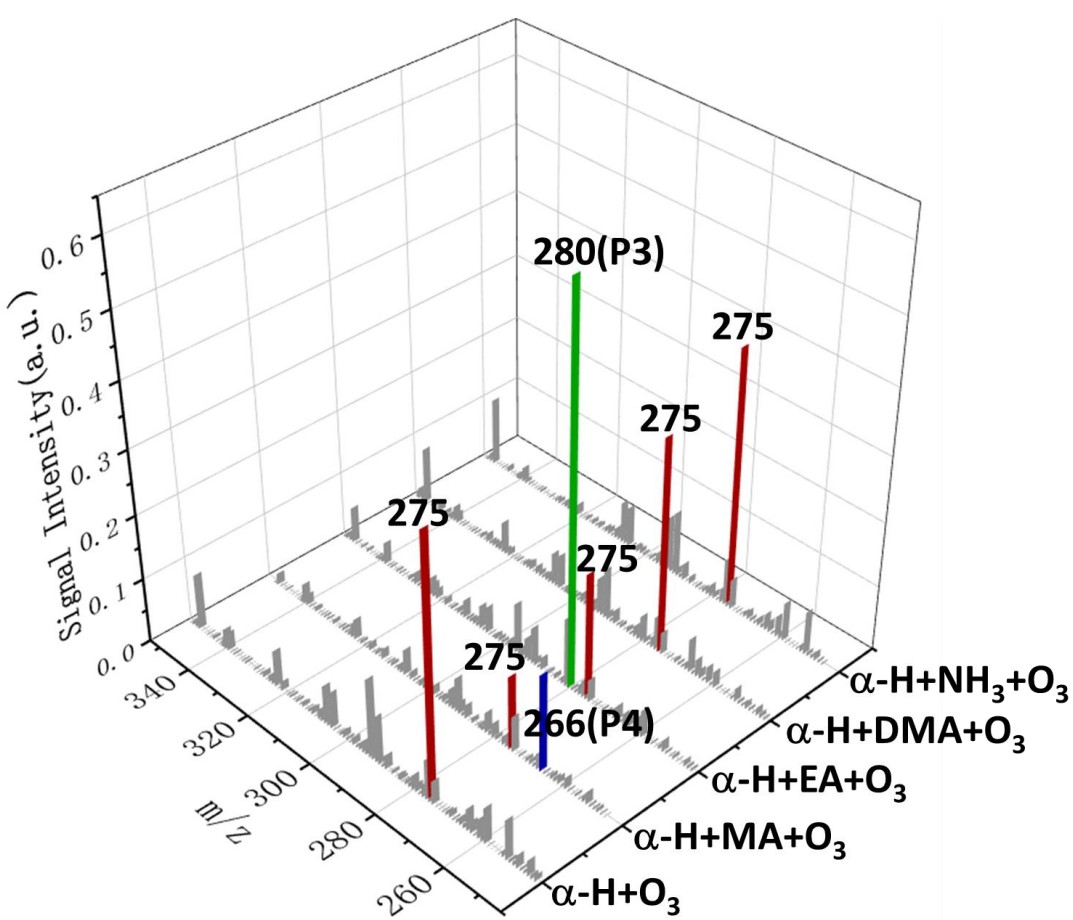

**Figure 4 Positive-ion ESI mass spectra of the products extracted in AN/W (vol/vol = 4/1) from ozonolysis of α-H in the absence/presence of MA, EA, DMA, or ammonia**




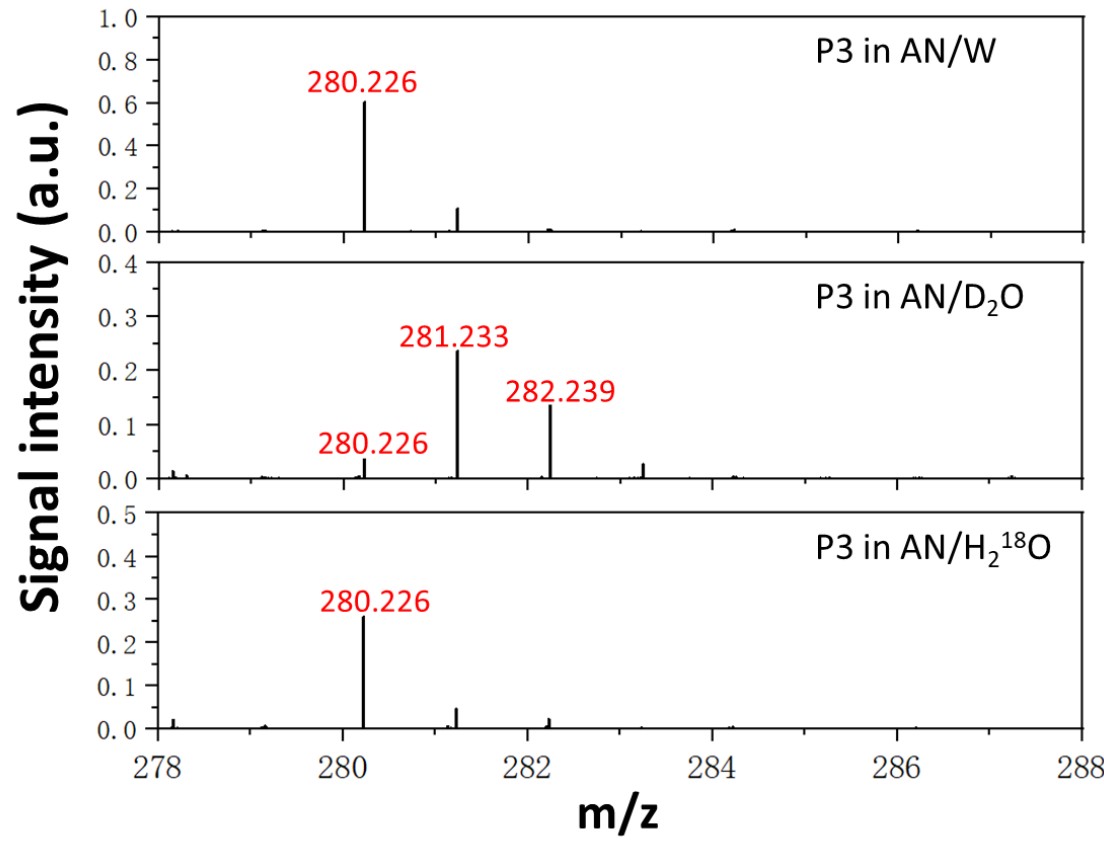

**Figure 5 High-resolution positive-ion ESI mass spectra of P3 extracted in AN/W (vol/vol = 4/1), AN/D₂O (vol/vol = 4/1) and AN/H₂¹⁸O (vol/vol = 4/1) solutions**



**Scheme 1 Major isomerized products of C₁₅H₂₄O₃ generated from the ozonolysis of β-C**





485    **Scheme 1 Major isomerized products of C₁₅H₂₄O₃ generated from the ozonolysis of β-C**





**Scheme 2 Possible structure of P1 generated from the reaction of β-C SOZ with EA**

β-C SOZ
($C_{15}H_{24}O_3$, MW204)

**P1**, m/z 280
($C_{17}H_{29}O_2N$ MW279)

490



**Scheme 3 Possible structure of P3 generated from the reaction of α-H SOZ with EA**

α-H SOZ
($C_{15}H_{24}O_3$, MW204)

**P3**, m/z 280
($C_{17}H_{29}O_2N$ MW279)