# Peer review of "A Possible Unaccounted Source of Nitrogen-Containing Compounds Formation in Aerosols: Amines Reacting with Secondary Ozonides"

_EGUsphere, 2023_

## Author Response (AR1)

**Manuscript ID: egusphere-2023-2080**

**Title:** A Possible Unaccounted Source of Nitrogen-Containing Compounds Formation in Aerosols: Amines Reacting with Secondary Ozonides

**The corresponding authors:** Prof. Taicheng An

**Dear Anonymous Referees,**

Thank you for the helpful and valuable review and comment. We have made careful revisions on the original manuscript according to your kind and helpful comments. The changed sentences have been marked as red color in the revised manuscript. Below is our point-by-point response to your comments:

**Anonymous Referee #1**

This manuscript by Qiu et al. mainly represents a laboratory study on the formation of N-containing compounds arising from a hitherto unreported route that SOZs react with amines. The topic of the study is highly relevant with journal and very timely, as N-containing compounds and SOZs both represent under-studied areas within the field of atmospheric chemistry, within the scope of ACP. The experiments were well designed with caution, and the analyses were performed in-depth. In particular, I think the method employing isotopically labelled water ($D_2O$ and $H_2^{18}O$) is an elegant way to reveal the structures of products. Thus, I recommend it can be published after following minor comments.

Detailed Comments:

1. The authors only presented the products less than m/z 350, however based on previous studies, numerous products from the ozonolysis of β-C or α-H lager than m/z 350 were also observed. Have the authors considered their interactions with amines?

**Author reply:** Thanks to the reviewer's kind reminder. The aim of this research is to explore the interactions of SOZs with amines. We selected β-C and α-H in this work, not only because they are representative sesquiterpenes, but SOZs are dominant products from their reactions with ozone.

In line 156-159, we revised as follows. "Because the aim of this study is to investigate the interactions of SOZs with amines, we only present a part of the products related to this study in mass spectra. The larger products from the ozonolysis of β-C as well as their potential interactions with amines will not be discussed in this work.".

2. The authors mentioned that SOZs are detected as $Na^+$-adducted, but the products arising from SOZs reacting with amines are detected as $H^+$-adducted. Could the authors explain this more explicitly, especially on where $Na^+$ came from, and if $Na^+$ is appropriate for SOZs detection.

**Author reply:** Thanks to the reviewer's kind reminder. In line 150-153, we revised as follows. "$Na^+$ was leached into the solution from trace metals in the laboratory glassware (Greaves and Roboz, 2014). $Na^+$ has an affinity toward an O-atom of a species possessing R-O-R' such as ethers (Sugimura et al., 2015). In addition, a previous study reported that SOZ originated from α-terpineol was detectable as $Na^+$-adducted species (Qiu et al., 2022).".

3. I noticed that this work was conducted at a low humidity in a smog chamber. Why the authors didn't perform experiments under different water vapor concentrations?

**Author reply:** Thanks to the reviewer's kind reminder. In this work, we are focused on the interactions of SOZs with amines. How water vapor affects their reaction seems to be an interesting topic, and we are considering that in our next work.

In line156-160, we revised as follows. "Because the aim of this study is to investigate the interactions of SOZs with amines, we only present a part of the products related to this study in

mass spectra. The larger products from the ozonolysis of β-C as well as their potential interactions with amines will not be discussed in this work. For the same reason, all the experiments were carried out at an extremely low humidity (RH ≤ 5%), to avoid the generation of unwanted products in the presence of high concentration water vapor (Kundu et al., 2017).".

4. SOZs are derived from Criegee intermediates which are highly reactive species. The authors should clarify the products detected are not from the reactions between Criegee intermediates and amines.

**Author reply:** Thanks to the reviewer's kind reminder. In line 164-165 we revised as follows.

"The results in Figure S4 provides another information that P1 is not the product from EA reacting with Criegee intermediate, the precursor of SOZ, which is a highly reactive species, because Criegee intermediate could not have survived for so long.".

**Anonymous Referee #2**

This study conducted chamber study to investigate the reactivities of secondary ozonides (SOZs) toward amines, and the target SOZs were produced by the ozonolysis of β-caryophyllene (-C) and α-humulene (α-H), both of which are representative sesquiterpenes and SOA markers. They found that the concerned SOZs readily reacts with ethylamine (EA) and methylamine (MA), but has inert reactivities toward dimethylamine (DMA) and ammonia. Interestingly, further isotope labelling experiments revealed that the products possessed different functional groups with the same amine, although β-C SOZ and α-H SOZ are isomerized species and share a same chemical structure. The manuscript present nice experimental results for exploring the formation pathways of N-containing compounds in atmospheric aerosols. The overall quality of the text is commendable, and I have several questions that need clarification before considering acceptance.

General Concerns/Weaknesses:

1. Introduction, there is limited information about the Nitrogen (N)-containing compounds, what and how it been acting as light absorbing species in aerosols, and which species of it would be hazardous to human health.

**Author reply:** Thanks to the reviewer's kind reminder. In line 26-32 we revised as follows. "The compounds possessing some N-containing functional groups, such as imidazole, pyrazine, and organonitrate groups, are recognized as acting as chromophores, which are closely associated with optical properties of the aerosols (Laskin et al., 2015; Moise et al., 2015). Meanwhile, some N-containing compounds are considered to be hazardous to human health. For instance, nitration of polycyclic aromatic compounds (PAHs) typically contributes more to the toxicity of ambient PM than parent PAHs (Albinet et al., 2008). Another study revealed that the compounds in some pollens will have a higher allergenic potential after being converted into N-containing species (Cuinica et al., 2014).".

2. Experimental, two extremely same Teflon reactors were mounted inside the smog chamber, but it seems All the experiments were carried out in one reactor, so why two reactors were designed?

**Author reply:** Thanks to the reviewer's kind reminder. This smog chamber was designed for the research that two photochemical experiments are needed to conducted at a same time (Please check the papers cited in line 89). In this study, it is unnecessary to monitor two experiments at a same time, thus one reactor is sufficient. In addition, that all the experiments are conducted in a same reactor may exclude some unexpected errors.

In line 92-93 we revised as follow. "In this study each experiment needs to be repeated at least 3 times. To avoid unexpected errors in the experiment processes, all the experiments were conducted in the same reactor.".

3. Line 91-98, about experimental procedure for the ozonolysis of β-C in the presence of EA, as the initial concentration of β-C/α-H (200 ppb) is times of $O_3$ (50 ppb), and $O_3$ will almost be

consumed intermediately, it would be challenging to keep the concentration of $O_3$ (50 ppb) in the reactor. I note that the concentrations of $O_3$ inside the reactor were monitored by an $O_3$ analyzer (Model 49i, Thermo Scientific).

**Author reply:** Thanks to the reviewer's kind reminder. The concentration of $O_3$ inside the reactor was calculated by power of ozone generator and injection time. Before of operation of ozonolysis experiment, the concentration of $O_3$ was confirmed instead of being monitored throughout the experiment.

We revised as follows in line 119-122. "$O_3$ was generated by a commercial ozone generator, and the amount of $O_3$ was carefully calculated according to the injection time and the power of the ozone generator. Before the operation of each experiment, $O_3$ was injected into the reactor filled with 1000 L zero air, and confirmed by an $O_3$ analyzer (Model 49i, Thermo Scientific).".

4. Line 107-108, it is unclear for the purpose to conduct the isotope labelling experiments here and how to obtain the chemical identification of the products. More explanation suggested to be added.

**Author reply:** Thanks to the reviewer's kind reminder. We revised in line 146-149 as follows. "$D_2O$ was used to test if a molecule contains an active H-atom. When a molecule possessing an active H-atom is dissolved in $D_2O$ solution, the active H-atom readily exchanges with D-atom of $D_2O$, increasing its molecular weight by 1 unit. The results in Figure S3 show that the product appeared at m/z 275 possesses no exchangeable H-atom, implying it should be $[SOZ + Na]^+$.".

Additionally, we revised in line 176-180 as follows. "As mentioned before, $D_2O$ was used to confirm if a molecule contains an active H-atom, while $H_2^{18}O$ was used to test if a molecule is carrying a carbonyl group. After being dissolved in $H_2^{18}O$ solution, the compound possessing a carbonyl group forms a gem-diol via the addition of $H_2^{18}O$ which is a reversible process. Since the concentration of $H_2^{18}O$ is overwhelming, $^{16}O$ of the compound will be almost replaced by $^{18}O$ of $H_2^{18}O$, resulting in 2 units rise in molecular weight.".

5. Line 149-150, it seems another experiment was conducted concerning on the reaction of EA in β-C SOZ solution. More details on this experiment should be provided. In addition, the above reaction occurred in the extraction process would be different with that in the pure liquid phase, so the possibility seems could not be ruled out.

**Author reply**: Thanks to the reviewer's kind reminder. The purpose of this experiment is to exclude the possibility that SOZ react with EA in the extraction process, in order to confirm the reaction of SOZ with EA indeed take place in the smog chamber.

To make it more explicitly, we revised in line 166-171 as follows. "Moreover, the particles generated from the ozonolysis of β-C in the absence of EA were sampled and dissolved in AN/W solution. EA was directly added into the solution. 30 min later the solution was analyzed by the mass spectrometer. As a result, no signal appeared at m/z 280 in mass spectra, indicating that β-C SOZ reacting with EA in liquid phase is not available. Through this experiment, the possibility that EA vapor condensed on the particles first and then react with β-C SOZ in the extraction process was ruled out, and it was determined that the reaction of β-C SOZ with EA occurred in the smog chamber.".